# Integrative transcriptomic analysis of tissue-specific metabolic crosstalk after myocardial infarction

Muhammad Arif[1†], Martina Klevstig[2†], Rui Benfeitas[3], Stephen Doran[4], Hasan Turkez[5], Mathias Uhlén[1], Maryam Clausen[6], Johannes Wikström[7], Damla Etal[6], Cheng Zhang[1], Malin Levin[2], Adil Mardinoglu[1,4]*, Jan Boren[2]*

[1]Science for Life Laboratory, KTH - Royal Institute of Technology, Stockholm, Sweden; [2]Department of Molecular and Clinical Medicine, University of Gothenburg, The Wallenberg Laboratory, Sahlgrenska University Hospital, Gothenburg, Sweden; [3]National Bioinformatics Infrastructure Sweden (NBIS), Science for Life Laboratory, Department of Biochemistry and Biophysics, Stockholm University, Stockholm, Sweden; [4]Centre for Host-Microbiome Interactions, Faculty of Dentistry, Oral & Craniofacial Sciences, King's College London, London, United Kingdom; [5]Department of Medical Biology, Faculty of Medicine, Atatürk University, Erzurum, Turkey; [6]Translational Genomics, BioPharmaceuticals R&D, Discovery Sciences, AstraZeneca, Gothenburg, Sweden; [7]Bioscience Cardiovascular, Research and Early Development, Cardiovascular, Renal and Metabolism (CVRM), BioPharmaceuticals R&D, AstraZeneca, Gothenburg, Sweden

**Abstract** Myocardial infarction (MI) promotes a range of systemic effects, many of which are unknown. Here, we investigated the alterations associated with MI progression in heart and other metabolically active tissues (liver, skeletal muscle, and adipose) in a mouse model of MI (induced by ligating the left ascending coronary artery) and sham-operated mice. We performed a genome-wide transcriptomic analysis on tissue samples obtained 6- and 24 hr post MI or sham operation. By generating tissue-specific biological networks, we observed: (1) dysregulation in multiple biological processes (including immune system, mitochondrial dysfunction, fatty-acid beta-oxidation, and RNA and protein processing) across multiple tissues post MI and (2) tissue-specific dysregulation in biological processes in liver and heart post MI. Finally, we validated our findings in two independent MI cohorts. Overall, our integrative analysis highlighted both common and specific biological responses to MI across a range of metabolically active tissues.

*For correspondence: adil.mardinoglu@kcl.ac.uk (AM); Jan.Boren@wlab.gu.se (JB)

†These authors contributed equally to this work

## Introduction

Cardiovascular disease (CVD) is the leading cause of death worldwide, accounting for more than 17 million deaths globally in 2016 (*WHO, 2019*). Myocardial infarction (MI) is one of the most common causes of CVD-related death and is the result of severe coronary artery disease that develops from tapered arteries or chronic blockage of the arteries caused by accumulation of cholesterol or plaque (atherosclerosis). Many behavioral risk factors (including unhealthy diet, physical inactivity, excessive use of alcohol, and tobacco consumption), which are responsible for hypertension, obesity, diabetes, and hyperlipidemia by significantly altering metabolism, are also implicated in MI. These abnormalities are known as the high-risk factors of MI and CVDs in general.

Systems biology has been used in many studies to reveal the underlying molecular mechanisms of complex human diseases and to answer important biological questions related to the progression,

**eLife digest** The human body is like a state-of-the-art car, where each part must work together with all the others. When a car breaks down, most of the time the problem is not isolated to only one part, as it is an interconnected system. Diseases in the human body can also have systemic effects, so it is important to study their implications throughout the body. Most studies of heart attacks focus on the direct impact on the heart and the cardiovascular system. Learning more about how heart attacks affect rest of the body may help scientists identify heart attacks early or create improved treatments.

Arif and Klevstig et al. show that heart attacks affect the metabolism throughout the body. In the experiments, mice underwent a procedure that mimics either a heart attack or a fake procedure. Then, Arif and Klevstig et al. compared the activity of genes in the heart, muscle, liver and fat tissue of the two groups of mice 6- and 24-hours after the operations. This revealed disruptions in the immune system, metabolism and the production of proteins. The experiments also showed that changes in the activity of four important genes are key to these changes. This suggests that this pattern of changes could be used as a way to identify heart attacks.

The experiments show that heart attacks have important effects throughout the body, especially on metabolism. These discoveries may help scientists learn more about the underlying biological processes and develop new treatments that prevent the harmful systemic effects of heart attacks and boost recovery.

diagnosis, and treatment of the diseases. The use of systems biology has aided the discovery of new therapeutic approaches in multiple diseases (*Mardinoglu et al., 2017a*; *Mardinoglu and Nielsen, 2015*; *Nielsen, 2017*) by identifying novel therapeutic agents and repositioning of existing drugs (*Turanli et al., 2019*). Systems biology has also been employed in the identification of novel bio-markers, characterization of patients, and stratification of heterogenous cancer patients (*Benfeitas et al., 2019*; *Bidkhori et al., 2018*; *Lee et al., 2016*). Specifically, integrated networks (INs) (*Lee et al., 2016*) and co-expression networks (CNs) (*Lee et al., 2017*) have been proven to be robust methods for revealing the key driver of metabolic abnormalities, discovering new therapy strategies, as well as gaining systematic understanding of diseases (*Bakhtiarizadeh et al., 2018*; *Mukund and Subramaniam, 2017*).

Previously, multiple studies in individual tissues have been performed and provided new insights into the underlying mechanisms of diseases (*Pedrotty et al., 2012*; *Das et al., 2019*; *Ounzain et al., 2015*; *Williams et al., 2018*). However, the crosstalk between different tissues and their dysregulation has not been examined in MI and other CVD-related complications (*Priest and Tontonoz, 2019*). Here, we performed an integrated analysis of heart and other metabolically active tissues (liver, skeletal muscle and adipose tissue) using a mouse model of MI. We used several systems biology approaches to obtain a systematic picture of the metabolic alterations that occur after an MI (*Figure 1A*), and validated our findings in two independent datasets.

## Results

### Differential expression analysis shows a pronounced effect on gene expression 24 hr post MI

To study global biological alterations and systemic whole-body effects associated with MI, we obtained heart, liver, skeletal muscle, and white adipose tissue from mice 6 hr and 24 hr after either an MI (induced by ligating the left ascending coronary artery) or a sham operation (as control). Total of 20 mice were used in this study (five mice in each time and condition combination) (*Figure 1A*). We generated transcriptomics data and identified differentially expressed genes (DEGs) 6 and 24 hr post MI and sham operation in all tissues, with the most significant differences occurring after 24 hr (*Supplementary file 1*, *Figure 1B*). Principal component analysis (PCA) showed a close clustering between the control (for both time points) and MI (6 hr and 24 hr separately) samples for heart tissue but clustering by extraction time points (6 hr and 24 hr clusters) for the other tissues (*Figure 1—figure supplement 1*). We present the transcriptional changes associated with MI in

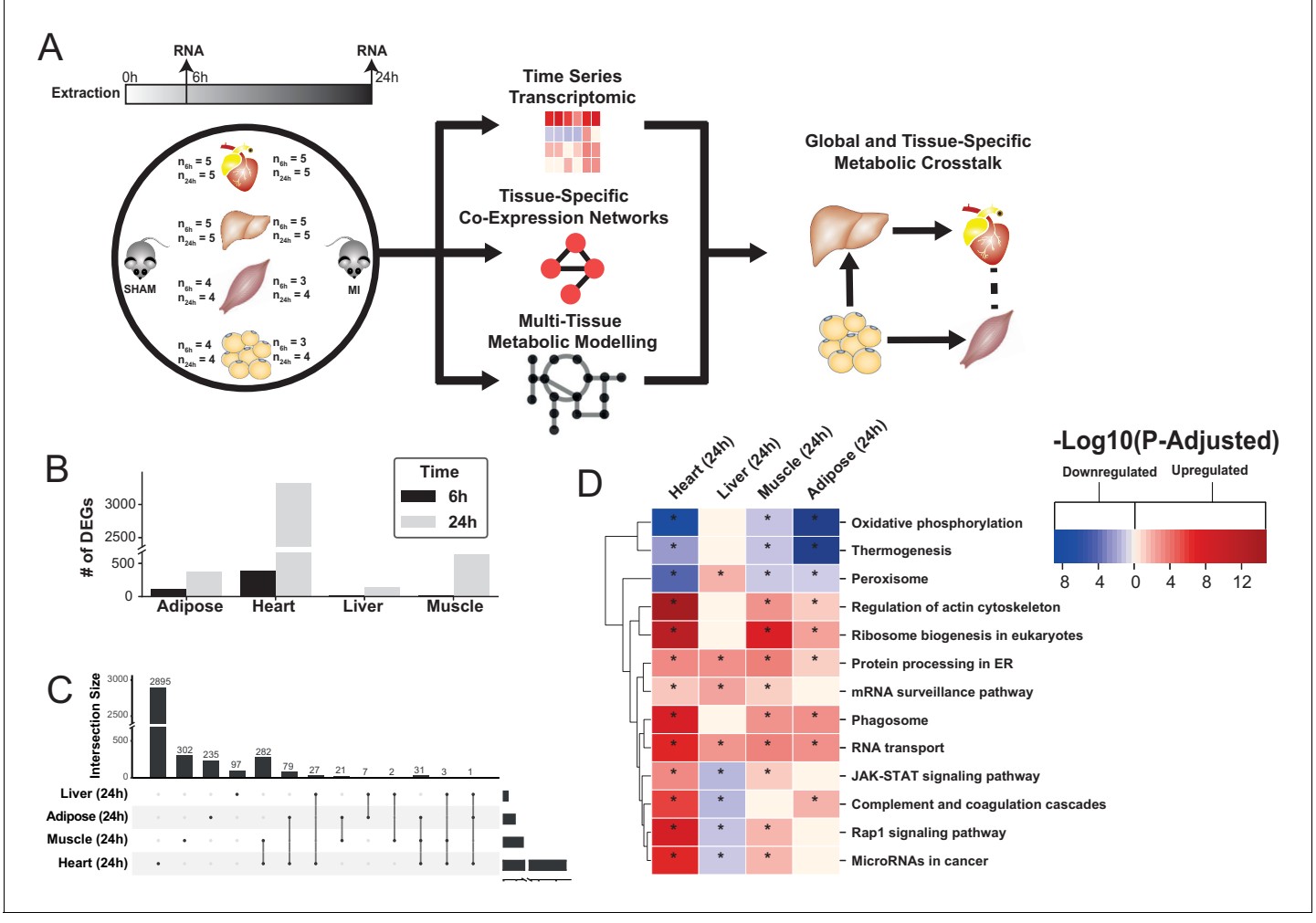

**Figure 1.** Study overview and transcriptional changes 24 hours after MI. (**A**) Overview of this study (**B**) Number of differentially expressed genes for each tissue at each time point. Effect of MI shown to be more pronounced after 24 hr. (**C**) UpSet plot to show intersection between differentially expressed genes (FDR < 5%) in different tissues. The plot showed that each tissue has its specific set of genes that were affected by MI. (**D**) KEGG pathway analysis (FDR < 0.05 in at least three tissues) for 24 hours post MI compared to its control for each tissue. We observed that 141 (5 upregulated) and 125 (14 upregulated) pathways are significantly altered in heart 6 and 24 hr after infarction, respectively. For other tissues, we found that 24 (9 upregulated), 61 (54 upregulated), and 48 (15 upregulated) pathways are altered in liver, muscle, and adipose, respectively.
The online version of this article includes the following figure supplement(s) for figure 1:

**Figure supplement 1.** Data exploration of the samples.
**Figure supplement 2.** KEGG pathway analysis results for Heart 6- and 24 hr post MI.
**Figure supplement 3.** KEGG pathway analysis results for each tissue liver, muscle, and adipose tissue 24 hr post MI.
**Figure supplement 4.** KEGG pathways related to cardiac problems show activation after an MI.

*Supplementary file 1* and the DEGs (FDR < 5%) using an UpSet plot (*Lex et al., 2014*) in *Figure 1C*.

All tissues showed a more pronounced effect in terms of the number of DEGs 24 hr post MI (*Figure 1C*). As expected, the most affected tissue was the heart (393 DEGs at 6 hr, 3318 DEGs at 24 hr, and 318 DEGs were the same at both time points). By contrast, 136, 641, and 374 genes were significantly changed in liver, skeletal muscle and adipose tissues 24 hr post MI compared to control, respectively. More than 33% of the DEGs that significantly changed in the other tissues also changed in the heart (*Figure 1C*). Interestingly, more than 97% of the shared DEGs between heart and skeletal muscle changed in the same direction, with corresponding numbers of 88% and 64% in adipose and liver, respectively.

# Functional analysis reveals widespread alterations of mitochondrial, fatty acid, immune, and protein and RNA-related biological processes post MI with liver shows contrasting trend

We performed gene-set enrichment analysis (GSEA) with KEGG pathways (*Supplementary file 2*, *Figure 1D*) and gene ontology (GO) biological processes (BPs) (*Supplementary file 3*, *Figure 2A*) to identify altered biological functions and pathways 24 hr after an MI. Mitochondrial functions (specifically, mitochondrial translation, respiratory chain and oxidative phosphorylation) were significantly downregulated in the heart, muscle and adipose tissues but not in the liver. Processes related to oxidative stress were upregulated in the heart and skeletal muscle. Fatty acid beta-oxidation was downregulated in the heart and adipose but upregulated in the liver. Processes and pathways related to immune systems were significantly upregulated in the heart and skeletal muscle but significantly downregulated in liver. Processes associated with protein and RNA processing, ribosome biogenesis

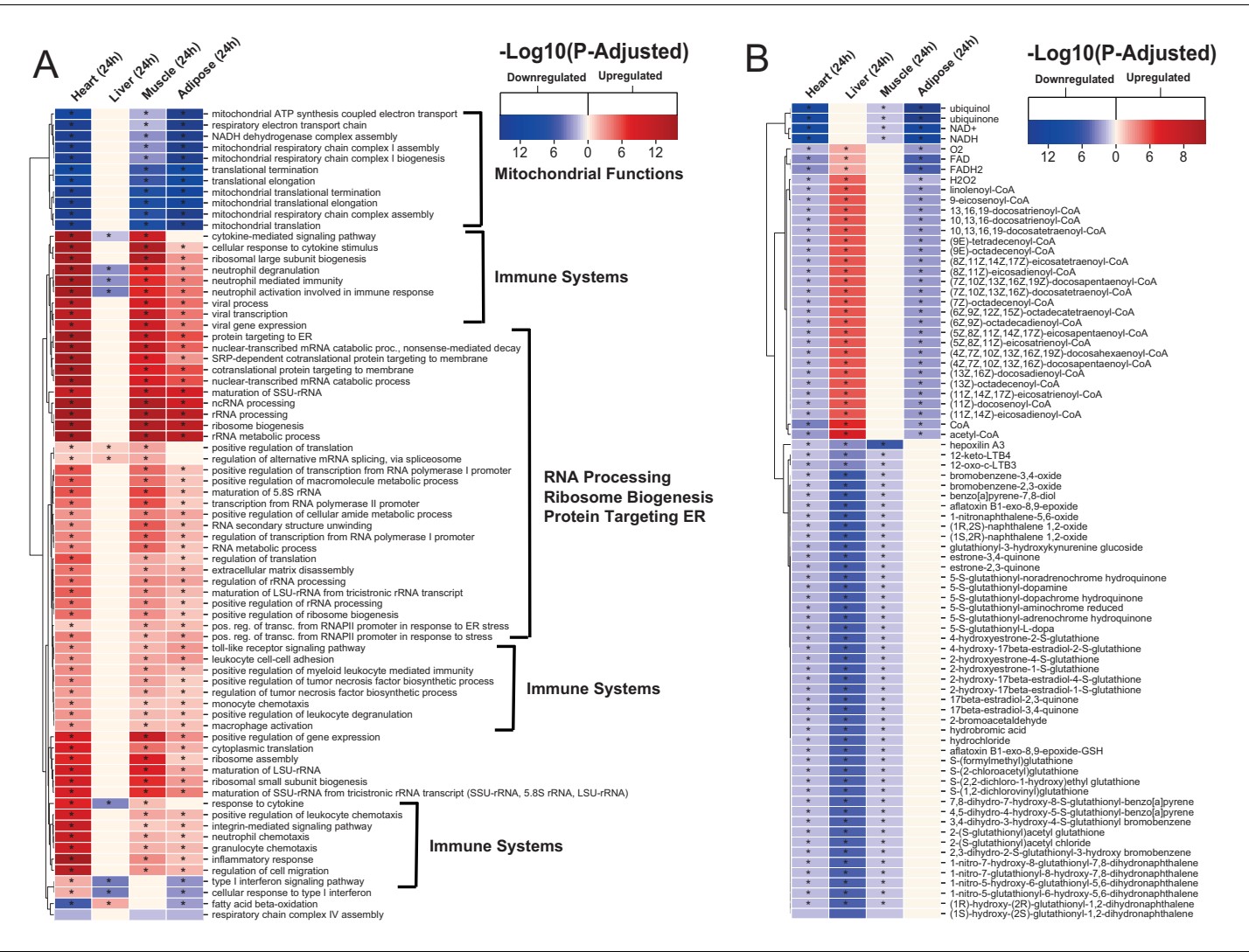

**Figure 2.** Gene ontology and reporter metabolites analysis results. (**A**) Functional analysis with GO (FDR < 0.05% in at least three tissues) revealed that 944 (919 upregulated) and 1019 (970 upregulation) BPs are significantly altered in heart 6 and 24 hr after infarction, respectively. The results also showed 38 (16 upregulated), 376 (357 upregulated), and 193 (116 upregulated) BPs are significantly altered 24 hr after infarction in liver, muscle and adipose, respectively. Most tissues show significant alterations in multiple biological processes, including mitochondrial functions, RNA processes, cell adhesion, ribosome, and immune systems. The results of this analysis showed alterations concordant with those observed for KEGG pathways. (**B**) Reporter metabolites analysis shows significant alteration in important metabolites. Our analysis revealed that 169, 324, 118, and 51 reporter metabolites are significantly altered in heart, liver, skeletal muscle and adipose tissues, respectively, at 24 hr post-infarction (Table S4).

and protein targeting endoplasmic reticulum were upregulated in all tissues except liver, whereas protein processing in endoplasmic reticulum and RNA transport pathways were upregulated in all tissues.

We also observed that liver was showing opposite trends compared to the other tissues in other important functions, such as fatty acid metabolism and immune response. By checking regulation at the gene level, we observed that only 16 DEGs in liver showed opposite regulation compared to the other tissues, whereas 97 out of the 136 DEGs in liver were not DEGs in any other tissues (*Supplementary file 4*). Therefore, the differences we observed in liver were mainly due to different DEGs rather than opposite regulation compared to other tissues.

## Tissue-specific altered biological functions point to specificity of metabolic and signaling responses to MI

The functional analysis also indicated that several metabolic pathways (including cholesterol, ascorbate and aldarate, linoleic acid, and sphingolipid metabolism pathways) and signaling pathways (including GnRH, FoxO, cAMP and prolactin signaling pathways) were significantly upregulated in heart 6 hr after an MI (*Supplementary file 2*, *Figure 1—figure supplement 2*). We also observed significant down regulation of tryptophan metabolism and upregulation of glycosaminoglycan biosynthesis in heart 24 hr after an MI (*Supplementary file 2*, *Figure 1—figure supplement 2*). Processes related to retinol metabolism were upregulated in heart at both timepoints. Pathways that were previously associated with cardiac hypertrophy and cardiac remodeling (e.g. JAK-STAT, MAPK, estrogen, and TNF signaling pathways, and ECM-receptor interaction) were significantly upregulated in heart 6 and 24 hr after an MI (*Figure 1—figure supplement 4*).

Our analysis also indicated significant metabolic differences in adipose tissue 24 hr after an MI (*Figure 1—figure supplement 3*). Fructose and mannose metabolism, glyoxylate and dicarboxylate metabolism, glycolysis/gluconeogenesis, and pentose phosphate pathways, glycine, serine and threonine metabolism and pyrimidine metabolism, as well as endocrine systems (e.g. insulin signaling pathway and regulation of lipolysis in adipocytes) were downregulated in adipose tissue.

We observed that the PPAR signaling pathway was upregulated, whereas glutathione was downregulated in liver 24 hr post-infarction (*Figure 1—figure supplement 3*). We found that sphingolipid metabolism and immune-related pathways were upregulated in skeletal muscle 24 hr post-infarction (*Figure 1—figure supplement 3*).

## Reporter metabolite analyses show significant alterations in fatty acid, amino acid, retinol, and estrogen metabolism post MI

To predict the effect of the transcriptional changes on metabolism, we performed reporter metabolite analyses (*Supplementary file 5*) using the gene-to-metabolites mapping from the Mouse Metabolic Reaction database (*Mardinoglu, 2015*); results in each tissue 24 hr after MI are shown in *Figure 2B*. In agreement with our analyses above, reporter metabolites related to oxidative phosphorylation, such as ubiquinol, ubiquinone, NADH and NAD+, were downregulated in all tissues except liver. Moreover, linolenoyl-CoA, acetyl CoA, and several other fatty acyl-CoA-related metabolites were downregulated in heart and adipose tissue but upregulated in liver. We also found that several 5-S-glutathionyl metabolite forms, known to be related to phenylalanine, tyrosine and tryptophan biosynthesis, were downregulated in heart, liver, and skeletal muscle. The same pattern of downregulation was also observed for metabolites related to estrogen metabolism, specifically metabolites related to oestrone and its glutathione conjugate derivative. Moreover, 12-keto-LTB4 and 12-oxo-c-LTB3, related to leukotriene metabolism, and hepoxilin A3, an arachidonic acid, were also found to be downregulated in heart, liver, and skeletal muscle.

The liver showed the highest alteration in reporter metabolites, which is attributed to its role as one of the most metabolically active tissues. We found that several reporter metabolites related to retinol metabolism, namely retinal, retinol, retinoate, and all-trans-18-hydroxyretinoic acid, were significantly downregulated only in liver tissue. Retinol metabolism has been previously associated with MI (*Lima et al., 2018*; *Palace et al., 1999*).

## Network analyses unveil universal and tissue-specific clusters and mechanisms post MI

The use of co-expression network (CN) analyses can assist in elucidating the functional relationships between genes in a specific cell and tissue (*Lee et al., 2017*). Here, we performed CN analysis to reveal the functional relationship between the DEGs by generating tissue-specific CNs and selected highly connected genes (the top 5% positively correlated genes that fulfilled FDR < 0.05) (*Table 1*). To better define the structure of the networks, we used the Leiden clustering algorithm (*Traag et al., 2019*) by maximizing the modularity scores (*Figure 3A–D*) and selected the clusters that include more than 30 genes. Next, we superimposed DEGs 24 hr post-infarction onto the network (*Supplementary file 1*) and identified the components of the clusters that were affected by an MI. We also used functional analysis with GO BP and KEGG pathways to understand the specific functions associated with each cluster by using the Enrichr algorithm (FDR < 0.05) (*Chen et al., 2013*; *Kuleshov et al., 2016*). We summarized the GO BP terms with Revigo (*Supplementary file 6*; *Supek et al., 2011*) and checked the average clustering coefficient to define the centrality of each cluster (*Supplementary file 6*; *Lee et al., 2017*). Among the clusters, we identified the key clusters as those with the highest average clustering coefficient, allowing us to identify sets of genes whose time-dependent coordinated changes showed the strongest relationships.

Interestingly, key clusters contained genes with similar functionalities including RNA processing, transports, and RNA metabolic processes in all tissue-specific CNs (*Supplementary file 6*). In addition, we found that the majority of the DEGs associated with those clusters were significantly upregulated. These observations strengthen the findings of the functional analysis above (*Figure 2A*) and further highlight how embryonically distinct tissues display similar functional responses to MI, with the most highly connected groups of genes preserved between different tissues (*Supplementary file 6*, *Figure 3E*).

## Community detection reveals tissue-specific clusters post MI

We investigated the tissue specificity of each cluster by performing enrichment analysis with data from the Mouse Gene Atlas (*Su et al., 2004*), which involved counting the number of tissue-specific genes.

The heart network showed the highest number of tissue-specific genes in cluster Heart-3 (302 genes). Based on DEG analysis, we found that 522 genes were downregulated and 192 genes were upregulated in the cluster. The enriched GO BP terms in the cluster were mitochondrial transport, protein processing and respiratory chain, cardiac muscle cell action potential, response to muscle stretch, and heart contraction (*Figure 3F*). We observed that the results of the KEGG pathway enrichment analysis were consistent with those obtained from GO BP analysis (*Supplementary file 6*).

In the liver network, cluster Liver-2 showed the highest tissue specificity (479 genes). In this cluster, we found that 15 genes were significantly downregulated and 17 genes were significantly upregulated. Based on GO BP enrichment analysis, the genes in this cluster were associated with cholesterol metabolism and homeostasis, lipid transport, glutathione metabolism, lipoprotein metabolism, and glucose 6-phosphate metabolism (*Supplementary file 6*). KEGG enrichment analysis also showed that the genes in the cluster were related to retinol, carbohydrate, lipid, and amino acid metabolism (*Supplementary file 6*).

The muscle network had two clusters with high tissue specificity: cluster Muscle-4 (276 genes) and Muscle-5 (143 genes). Muscle-4 showed association with GO BP terms such as mitochondrial transport, protein processing and respiratory chain, response to muscle stretch, and muscle contraction

**Table 1.** Properties of the co-expression network.

| Tissue | # of Genes | # of Edges | # of Clusters | Modularity scores |
|---|---|---|---|---|
| Heart | 8793 | 1570898 | 7 | 0.540 |
| Liver | 7760 | 1103589 | 6 | 0.577 |
| Muscle | 8834 | 1660603 | 7 | 0.521 |
| Adipose | 10790 | 2636378 | 8 | 0.495 |

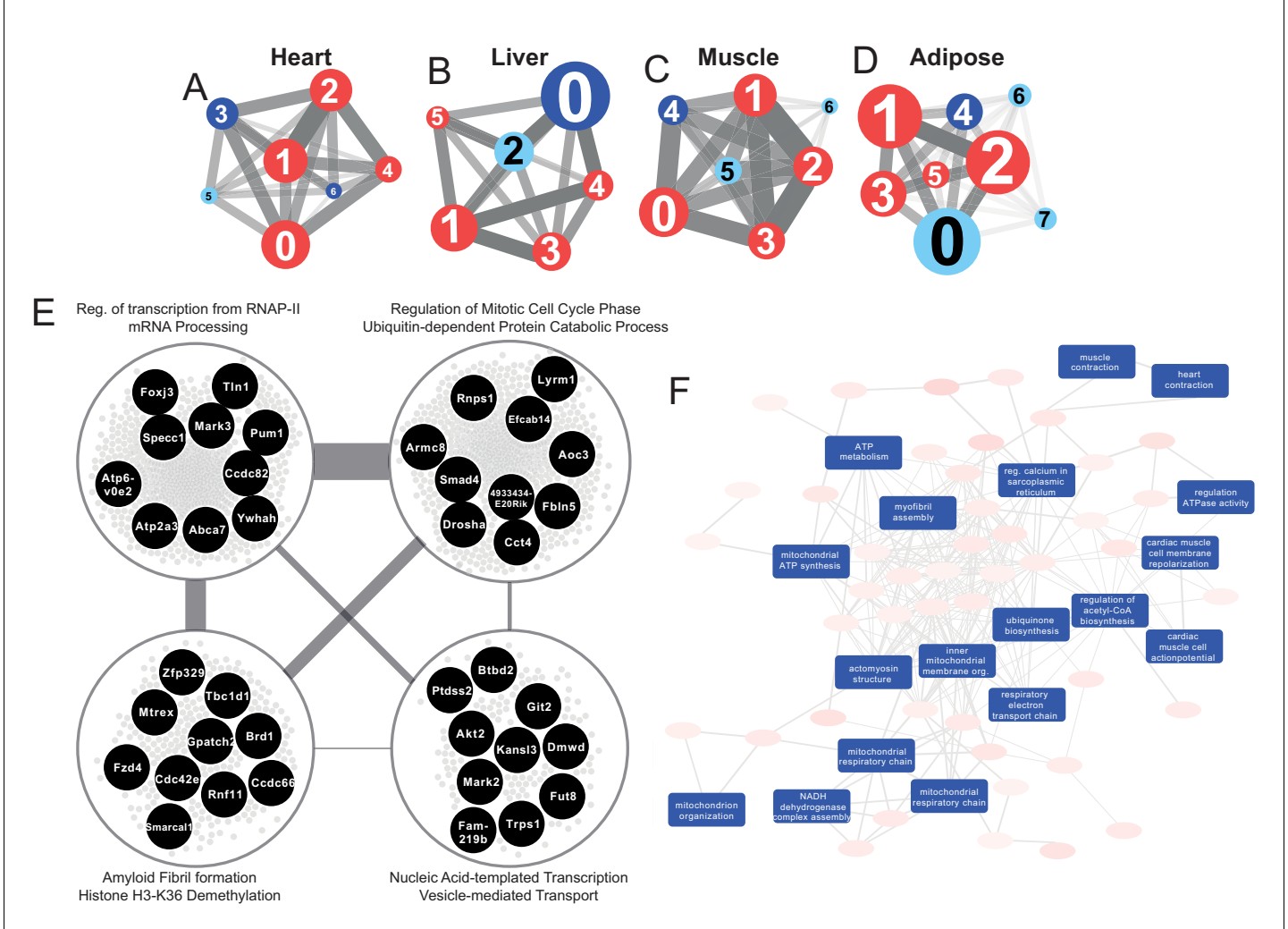

**Figure 3.** Tissue-specific gene co-expression network analyses. (**A**) Heart co-expression network clusters with superimposed DEGs 24 h post-infarction (Blue = downregulated, Red = upregulated) marked with the cluster numbers. The edges between the clusters were aggregation of the inter-cluster edges (**B**) Liver. (**C**) Muscle. (**D**) Adipose. (**E**) Intersection of the most central clusters in all tissues shows that the central architecture of the network was conserved in all tissues. We found four sub-clusters within the network intersection. Top 10 most connected genes are marked in black. (**F**) Enriched GO BP in heart-specific cluster generated by Revigo.

(*Supplementary file 6*). In contrast, the KEGG pathway in this cluster showed relation to glycolysis/ glucogenesis, propanoate metabolism, glyoxylate and dicarboxylate metabolism, and several signaling pathways (e.g. oxytocin, glucagon, cGMP-PKG, and HIF-1) (*Supplementary file 6*). Muscle-5 was enriched in GO BP terms associated with protein dephosphorylation, muscle contraction and intracellular protein transport (*Supplementary file 6*). We also found that insulin, MAPK and Wnt signaling pathways were associated to Muscle-5 from the KEGG enrichment analysis (*Supplementary file 6*).

The adipose tissue network showed tissue specificity in cluster Adipose-2 (33 genes), which is associated with GO BP processes including mRNA processing, regulation of mitotic cell cycle phase, ribosome biogenesis, and viral processes (*Supplementary file 6*). We observed that the results of the KEGG pathway enrichment analysis were consistent with those obtained from GO BP analysis, with additional associations with multiple signaling and regulatory pathways (*Supplementary file 6*).

## Tissue-specific clusters show important tissue-specific changes post MI

To understand the specific behavior of each tissue, we further studied the tissue-specific clusters in the CNs (*Figure 4A*). Heart specific cluster, Heart-3, was driven by several central genes including

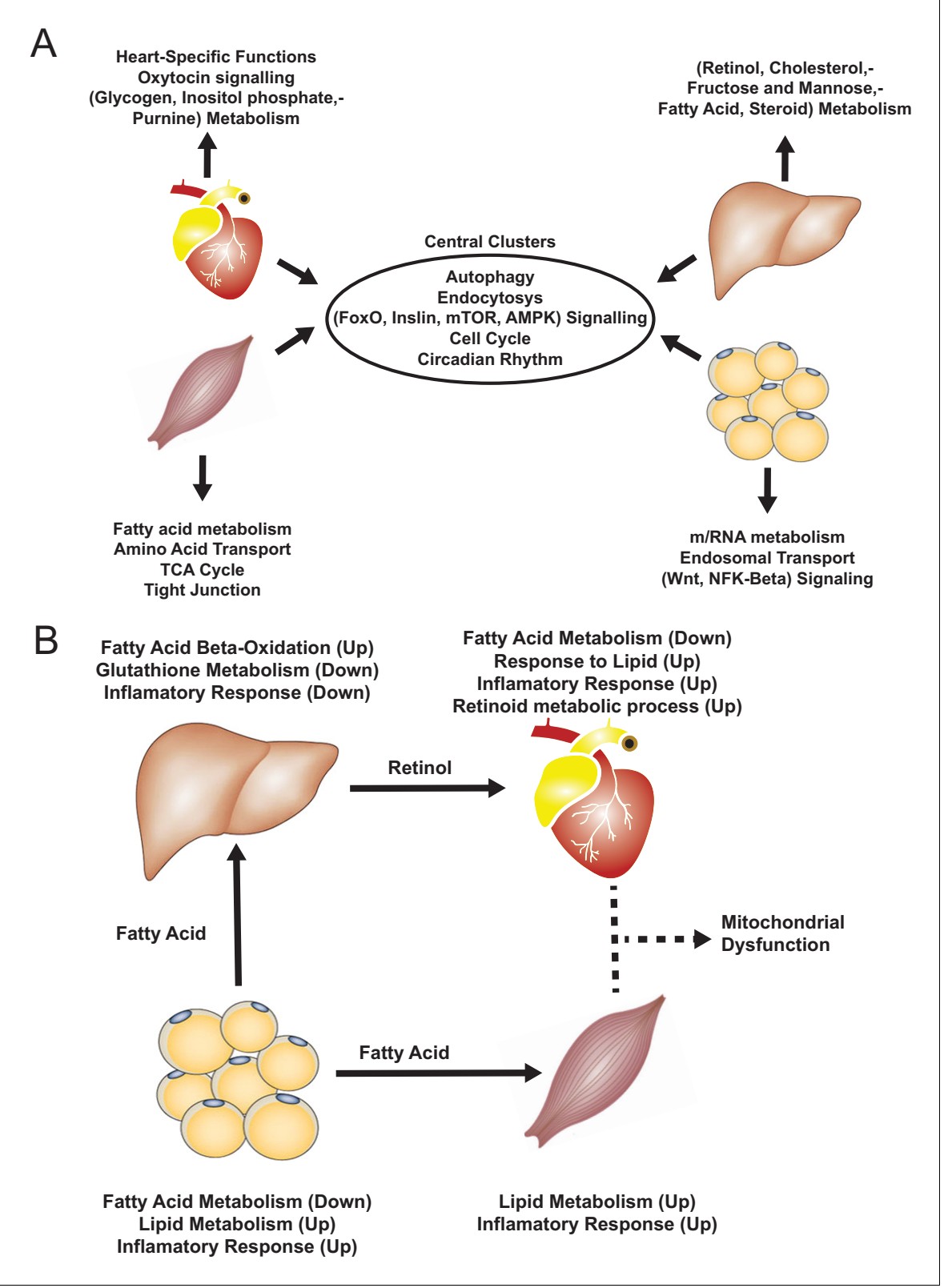

**Figure 4.** Functional analysis of network clusters and hypothesized metabolites flow. (**A**) Similarity of functions in the most central cluster and specific functions of each tissue-specific cluster. (**B**) Functional analysis for each tissue and hypothesized flow of metabolites.

The online version of this article includes the following figure supplement(s) for figure 4:

*Figure 4 continued on next page*

*Figure 4 continued*

**Figure supplement 1.** cGMP-PKG with overlay data from differential expression and reporter metabolites analysis.

**Figure supplement 2.** HIF-1 signaling pathway with overlay data from differential expression and reporter metabolites analysis.

*Pln*, *Pde4b*, and *Atp2a2* (related to regulation of cardiac muscle contraction) and *Pdha1* and *Vdac1* (related to mitochondrial functions). These genes were also found to be significantly differentially expressed in heart 24 hr post MI (*Supplementary file 1*). Genes in the heart-specific cluster were related to multiple other processes/pathways, for example oxytocin signaling pathway, and several metabolic pathways (glycogen, inositol phosphate, and purine) (*Supplementary file 6*).

Mitochondrial dysfunction in the heart leads to disturbance of energy (ATP) production (*Kiyuna et al., 2018*; *Palaniyandi et al., 2010*) and, in the presence of oxygen, to accumulation of reactive oxygen species (ROS), which can cause oxidative stress. *Vdac1*, a key gene for regulation of mitochondria function and one of the central genes in the heart-specific cluster (see above), is significantly downregulated in MI (*Camara et al., 2017*). *Vdac1* is located in the outer mitochondrial membrane and is involved directly in cardioprotection (*Schwertz et al., 2007*) within the cGMP/PKG pathway (*Figure 4—figure supplement 1*). In the same pathway, we also observed down-regulation of the reporter metabolite hydrogen peroxide (*Supplementary file 5*), a ROS that is related to cardioprotection (*Schwertz et al., 2007*; *Yada et al., 2006*). We also observed downregulation of *Pdha1*, which is known to have a substantial role in both the HIF-1 signaling pathway and the pyruvate metabolism pathway that converts pyruvate to acetyl-CoA in the mitochondria (*Figure 4—figure supplement 2*). Acetyl-CoA is used in the TCA cycle to produce NADH and FADH2, which are both needed for ATP production and were downregulated in our reporter metabolite analysis of the heart. Our findings are thus consistent with dysfunctional mitochondria and ATP production in the heart in response to an MI. *Pdha1* has been also been linked to the heart sensitivity during to ischemic stress, where its deficiency can compromise AMP-activated protein kinase activation (*Sun et al., 2016*).

In skeletal muscle and adipose tissue, we found that central genes in their respective tissue-specific clusters related to fatty acid metabolism and lipid metabolism were significantly altered (*Supplementary file 6*, *Figure 5*). In liver-specific cluster, we found that their central genes were related to fatty-acid beta oxidation (*Cyp4a31*, *Cyp4a32*) and glutathione metabolism (*Gstm3*) (*Supplementary file 6*, *Figure 5A*). Alterations of fatty acid beta-oxidation and glutathione metabolism have previously been reported in non-alcoholic fatty liver disease, a known risk factor of CVD (*Mardinoglu et al., 2017b*; *Alexander et al., 2019*). Moreover, in liver, we also found that retinol metabolism was uniquely related to genes in the liver-specific cluster, mainly driven by four significantly differentially expressed central genes of the clusters, that is *Cyp26a1*, *Cyp4a31*, *Cyp4a32*, and *Hsd17b6* (*Supplementary file 6*). A previous study showed that mortality from CVD in older individuals was accompanied by impaired liver ability to store retinol (*Lima et al., 2018*).

## Multi-tissue modeling reveals key metabolic pathways affected post MI

To investigate the metabolic responses to MI in and across tissues in the mice, we constructed a multi-tissue genome-scale metabolic model. The model consisted of five tissue-specific genome scale metabolic models, namely heart, liver, skeletal muscle, adipose, and small intestine. The small intestine model (for which we do not have transcriptomic data) was added to include ingestion and conversion of dietary nutrients into chylomicrons, which are directly secreted into blood and transport lipids to other tissues (*Mardinoglu, 2015*). The final mouse multi-tissue model included 19,859 reactions, 13,284 metabolites, 7116 genes, and 41 compartments. We predicted the metabolic fluxes in mice 24 hr after an MI or sham operation by integrating the dietary input, tissue-specific resting energy expenditure and transcriptomics data.

The modeling showed that oxygen uptake, carbon dioxide production and the oxidative phosphorylation pathway in heart, adipose and skeletal muscle were decreased in MI mice, in agreement with the downregulation of oxidative phosphorylation we observed in these tissues (*Supplementary file 7*). By contrast, liver showed slightly increased oxygen uptake, which might be due to the slightly (not statistically significant) upregulated oxidative phosphorylation (*Supplementary file 7*). These findings indicate that the changes in oxygen and carbon dioxide

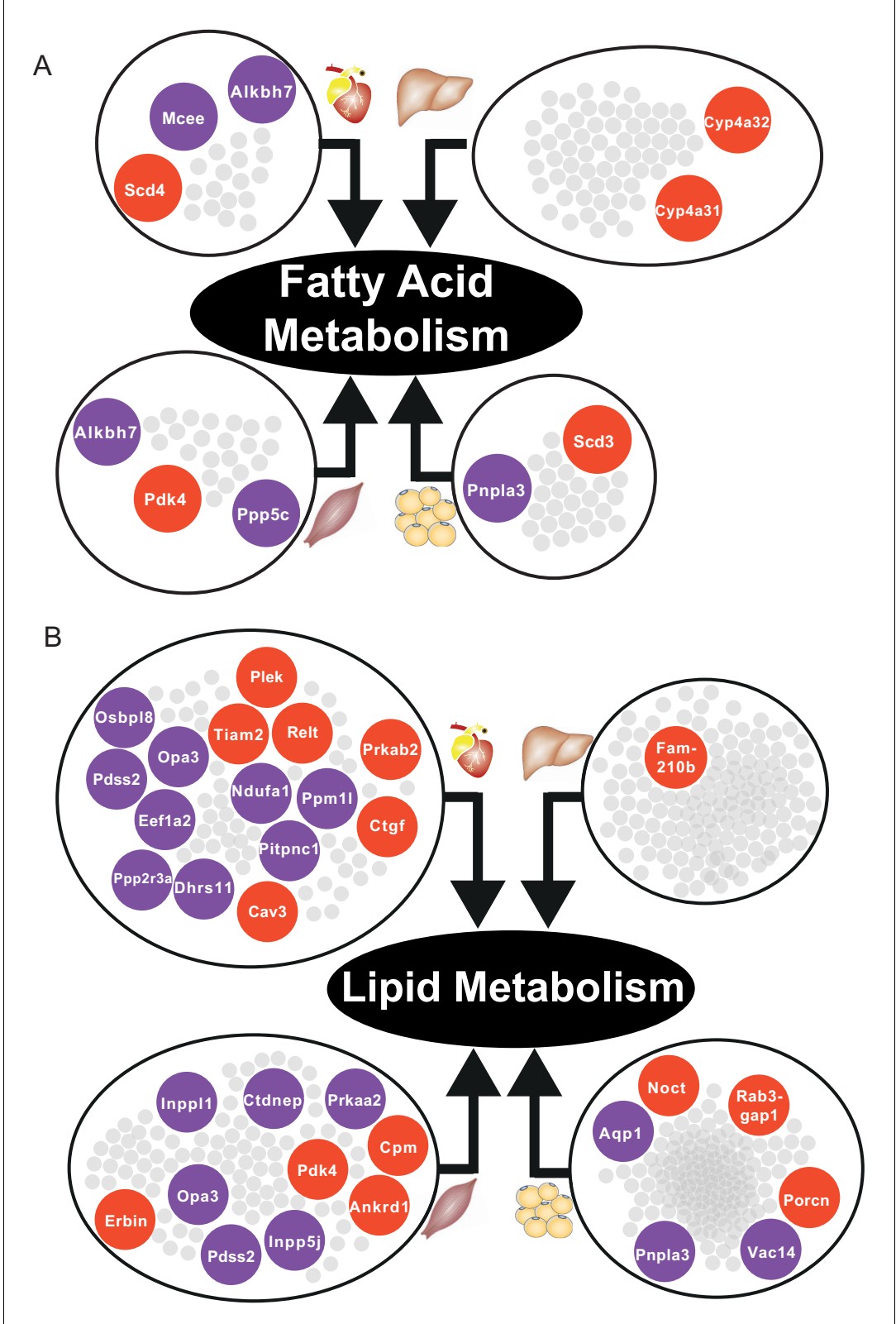

**Figure 5.** Central DEGs in fatty acid and lipid metabolism. (**A**) Significantly differentially expressed central genes of each tissue-specific cluster to fatty acid metabolism, as one of the most affected metabolic process. (**B**) Lipid metabolism. Red = upregulated, blue = downregulated.

fluxes and the oxidative phosphorylation pathway could serve as a positive control for predicting the changes due to MI in the fluxes.

Next, we investigated the tissue-specific metabolic flux changes in the same model (*Supplementary file 7*). We found that the pentose phosphate pathway was upregulated in heart 24 hr post MI, consistent with upregulated glucose metabolism after an MI. Elevated glycolysis could allow the heart to rapidly generate energy under stress conditions, and the enhanced pentose phosphate pathway could increase the NADPH level, which could help maintain the level of reduced glutathione in heart (*Tran and Wang, 2019*). In addition, we observed an increase uptake of alpha-ketoglutarate (AKG) of heart 24 hr after MI. It has been reported that supplementation of AKG could prevent heart from ischaemic injury (*Kjellman et al., 1995*), and the increased uptake of AKG we observed after MI might be a natural protective metabolic response to MI. Moreover, we found there is a net lactate metabolic flux coming from liver to heart in the MI group. The influx of lactate has been reported to be positively correlated with the fraction of regional ejection of heart (*Hattori et al., 1985*) and this net flux not only agrees well with the previous report but also additionally suggested the source of the lactate. We also found that adipose tissue secreted more ketone bodies, including acetoacetate and butyrate, into plasma; the plasma level of ketone bodies has been reported as a stress marker in acute MI (*Miyamoto et al., 1999*). Notably, relatively small metabolic changes were found in liver and skeletal muscle, which is probably due to the small number of transcriptomic changes in metabolic pathways in these tissues.

## Validating our findings with publicly available datasets

We validated our observations in heart tissue in two independent cohorts of bulk RNA-seq data from mouse heart (*Supplementary file 8*). We filtered both validation cohorts to get and analyzed only 24 hr post-MI data. We found that there were 2169 DEGs from our heart 24 hr post MI data were validated in at least one of the independent cohorts (959 DEGs validated in both) (*Figure 6A*). We also found that 109 out of the 123 most connected genes in our heart-specific cluster were also significantly differentially expressed in at least one of the independent cohorts (81 in both). By performing functional analysis of the validation cohorts, we found that ~61% of GO BP and 84% of KEGG pathways identified in our analysis of the heart were also present in at least one of the validation cohorts 24 hr after infarction (*Figure 6B–C*). In both cohorts, we observed downregulation of mitochondrial functions and fatty acid metabolism processes. We also observed upregulation of processes and pathways related to retinol metabolism and inflammatory response in both validation cohorts.

## Identification of driver genes in MI

We observed that *Flnc*, *Lgals3*, *Prkaca*, and *Pprc1* showed important role to MI. These genes were 4 of 16 genes that were DEGs in at least three tissues and validated in both validation cohorts

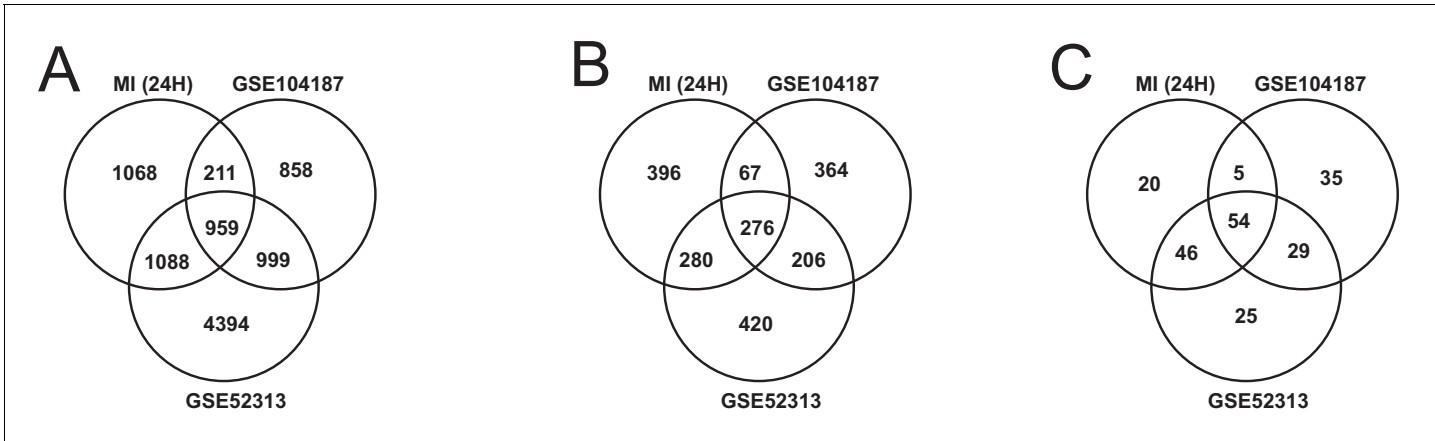

**Figure 6.** Comparison of our analysis results with the independent validation cohorts. (**A**) DEGs intersection of our data and validation cohort (**B**) and (**C**) intersection of functional analysis results (GO BP and KEGG Pathways) of our data and validation cohort.

(*Supplementary file 9*). *Flnc*, *Lgals3*, and *Pprc1* were upregulated in heart, skeletal muscle, and adipose, whereas *Prkaca* was downregulated in these three tissues. We further retrieved their neighbors at each tissue specific CNs, showed their regulations from differential expression results, and performed functional analysis in *Supplementary file 9*.

*Flnc*, which encodes filamin-C, was part of heart and skeletal muscle-specific CN cluster (Figure S4). Its neighbor genes were found to be significantly (FDR < 0.05) associated to several functions, including TCA cycle, pyruvate metabolism, glycolysis pathway, and involved in mitochondrial functions. Specifically, they were related to heart-specific processes in heart, VEGF signaling pathway in muscle, carbohydrate metabolism in adipose, and to MAPK signaling pathway and muscle contraction in heart and muscle.

*Lgals3* (encodes galectin-3) and *Prkaca* were among the most central genes in central clusters (*Supplementary file 6*). The neighbors of *Lgals3* were significantly related to cell cycle and protein digestion and absorption pathway in all tissues, and to RNA and mRNA related-processes in muscle and adipose tissue. The neighbors of *Prkaca* were related to insulin signaling pathway in heart and adipose, and several mitochondrial functions in adipose. *Pprc1* was part of most central clusters in heart and adipose tissue CN, and its neighbors were related to ribosomal RNA processing and ribosome biogenesis.

## Discussion

CVD has a complex etiology and is responsible for a range of systemic effects, hindering our understanding of its consequences on different tissues. Here, we took advantage of the technological advances in high-throughput RNA-seq and applied integrative network analyses to comprehensively explore the underlying biological effects of MI. Specifically, we generated RNA-seq data from heart, liver, skeletal muscle, and adipose tissue obtained from mice 6 and 24 hr after an MI or sham operation. We used transcriptomics data analyses (differential expression, functional analysis, and reporter metabolites analysis) to determine the systemic effects of the MI across multiple tissues. Moreover, we performed CN analyses to pinpoint important key and tissue-specific clusters in each tissue, and identified the key genes in each cluster. Finally, we used a whole-body modeling approach to identify the crosstalk between tissues and reveal the global metabolic alterations, before finally validating our findings with publicly available independent MI cohorts.

Based on our analyses, we observed downregulation of heart-specific functions and upregulation of lipid metabolism and inflammatory response in heart, muscle, and adipose tissue after an MI (*Figure 4B*). Liver showed a distinct response with respect to the other three tissues, including downregulation of inflammatory response. We observed that fatty acid metabolism was downregulated in heart and adipose tissue, whereas fatty acid beta-oxidation was upregulated and glutathione metabolism was downregulated in liver. We also observed upregulation of oxidative stress in heart and skeletal muscle. We also observed downregulation of mitochondrial functions in heart, muscle, and adipose tissue. Furthermore, we found upregulation of retinol metabolism in heart and downregulation of retinol metabolites in liver (*Figure 4B*).

We hypothesized that downregulation of fatty acid metabolism from adipose tissue was due to exchange of fatty acids with other tissues (liver and muscle) (*Figure 4B*). We also observed the flow of retinol from liver to heart during MI, consistent with previous reports (*Palace et al., 1999*). These MI-associated alterations lead to dysfunctional mitochondria and decreased energy production, especially in heart and skeletal muscle.

We also validated our results with publicly available MI datasets generated in separate independent studies. The validation results strengthened our findings on the altered functions/pathways and the important heart-specific genes after an MI.

Importantly, our analyses of gene clusters highlighted multiple key genes in the response to MI in different tissues. Specifically, we observed that *Flnc*, *Prkaca*, *Lgals3*, and *Pprc1* showed important responses in heart, skeletal muscle, and adipose tissue. *Flnc* is involved in actin cytoskeleton organization in heart and skeletal muscle, and previous studies have shown that this gene has critical role in CVD (*Zhou et al., 2020*; *Hall et al., 2020*). Similarly, *Prkaca*, an important metabolic gene, has also been shown to play an important function during CVD (*Diviani et al., 2011*; *Turnham and Scott, 2016*; *Bers, 2008*). *Lgals3*, related to acute inflammation response, has been studied intensively in recent years as a key gene in CVD, and as a potential CVD therapy target (*Zhong et al.,*

*2019*; *Suthahar et al., 2018*). Lastly, *Pprc1*, as important regulator of mitochondrial biogenesis, has not been explored for its direct relationship with CVD; however, mitochondrial biogenesis appears to be an important response to CVD (*Ren et al., 2010*; *Siasos et al., 2018*; *Piantadosi and Suliman, 2012*).

We recognized several limitations to be noted on this research. First, only transcriptomic data was analyzed in this research, hence the sensitivity might be limited especially for short timepoint, for example 6 hr after MI. Second, we focused our analysis in this research only on protein-coding genes. Third, to explore more about the shift in metabolism due to MI, longer timepoints needs to be explored. This opens new opportunities for future research, including analyzing the non-protein-coding gene signatures and longer timepoints.

In summary, we systematically unveiled the deregulation of biological processes and pathways that resulted from MI in heart, liver, muscle, and adipose tissue by integrating transcriptomic data and the use of biological networks. We also identified the key clusters and central genes using generated tissue-specific CNs. In this study, we demonstrated a strategy to utilize multi-tissue transcriptomic data to identify alteration of biological processes and pathways to systemically explore the effect of a disease.

# Materials and methods

**Key resources table**

| Reagent type (species) or resource | Designation | Source or reference | Identifiers | Additional information |
|---|---|---|---|---|
| Commercial assay or kit | RNeasy Fibrous Tissue Mini Kit | Qiagen | | Heart and Skeletal Muscle Tissue |
| Commercial assay or kit | RNeasy Mini Kit | Qiagen | | Liver Tissue |
| Commercial assay or kit | RNeasy Lipid Tissue Mini Kit | Qiagen | | Adipose Tissue |
| Commercial assay or kit | cDNA Reverse Transcription Kit | Applied Biosystems | | |
| Commercial assay or kit | TaqMan real-time PCR in a ViiA seven system | Applied Biosystems | | |
| Commercial assay or kit | NovaSeq6000 | Illumina | | |
| Software, algorithm | NovaSeq Control Software 1.6.0/RNA v3.4.4 | Illumina | | |
| Software, algorithm | CASAVA Software Suite | Illumina | | |
| Software, algorithm | Kallisto | | RRID:SCR_016582 | |
| Software, algorithm | Python 3.7 | Python Programming Language | RRID:SCR_008394 | |
| Software, algorithm | sklearn | Python Package | RRID:SCR_019053 | |
| Software, algorithm | R | R Project for Statistical Computing | RRID:SCR_001905 | |
| Software, algorithm | Rpy2 | Python Package | https://rpy2.github.io/ | |
| Software, algorithm | DESeq2 | R Package | RRID:SCR_015687 | |
| Software, algorithm | PIANO | R Package | RRID:SCR_003200 | |
| Software, algorithm | SciPy | Python Package | RRID:SCR_008058 | |
| Software, algorithm | Statsmodel | Python Package | RRID:SCR_016074 | |
| Software, algorithm | iGraph | Python Package | RRID:SCR_019225 | |
| Software, algorithm | Leiden Clustering | Python Package | https://github.com/vtraag/leidenalg | |
| Software, algorithm | Matlab | Mathworks | RRID:SCR_001622 | |

## Induction of MI

Ten-week-old male C57Bl/6N mice were fasted for 4 hr before induction of myocardial infarction. The mice were then anesthetized with isoflurane, orally intubated, and connected to a small-animal ventilator (SAR-830, Geneq, Montreal, Canada) distributing a mixture of oxygen, air and 2–3% iso-flurane. ECG electrodes were placed on the extremities, and cardiac rhythm was monitored during surgery. An incision was made between the 4th and 5th ribs to reveal the upper part of the anterior left ventricle (LV) wall and the lower part of the left atrium. Myocardial infarction was induced by ligating the left anterior descending (LAD) coronary artery immediately after the bifurcation of the left coronary artery 1. The efficacy of the procedure was immediately verified by characteristic ECG changes, and akinesis of the LV anterior wall. After verification of the infarction, the lungs were hyperinflated, positive end-expiratory pressure was applied, and the chest was closed. Sham mice were handled identically (fasted, anesthetized, intubated, and connected to ventilator, and subse-quently incised between 4th and 5th ribs), but no ligation of the LAD coronary artery was performed (and thus, no ischemia was induced in these mice). The mice received an intraperitoneal injection of 0.1 ml buprenorphine to relieve postoperative pain and were allowed to recover spontaneously after stopping isoflurane administration. Mice were killed with an overdose of isoflurane 6 hr or 24 hr after occlusion or sham operation. We collected the left ventricle (the whole left ventricle containing mainly infarcted tissue) of the heart, whereas white adipose tissue (WAT) was collected from the abdomen and musculus soleus was taken as the muscle tissue. Mouse hearts and biopsies from the liver, muscle and WAT were snap-frozen in liquid nitrogen and stored at −80°C until analysis. All mice studies were approved by the local animal ethics committee and conform to the guidelines from Directive 2010/63/EU of the European Parliament on the protection of animals used for scien-tific purposes.

## Echocardiography in mice

Echocardiographic examination, using VisualSonics VEVO 2100 system (VisualSonics Inc, Ontario, Canada), which includes an integrated rail system for consistent positioning of the ultrasound probe was performed 6 and 24 hr after an MI to determine the size of the MI. We calculated infarct size based on wall motion score index (WMSI) 24 hr after myocardial infarction by a 16-segments model on three short axis images, as 0 for normal, ½ for reduced wall thickening and excursion in a seg-ment and one for no wall thickening and excursion in a segment. WMSI was calculated as the sum of scores divided by the total number of segments. Hair removal gel was applied to isofluorane-anes-thetized (1.2%) mice chest to minimize resistance to ultrasonic beam transmission. The mice were then placed on a heating pad and extremities were connected to an ECG. A 55 MHz linear trans-ducer (MS550D) was used for imaging. An optimal parasternal long axis (LAX) cine loop of >1000 frames/s was acquired using the ECG-gated kilohertz visualization technique. Parasternal short axis cine-loops were acquired at 1, 3, and 5 mm below the mitral annulus. Infarct size was calculated based on wall motion score index 6 and 24 hr after myocardial infarction by a 16-segments model on LAX and three short axis images view, as 0 for normal, ½ for reduced wall thickening and excur-sion in a segment and one for no wall thickening and excursion in a segment. The data were evalu-ated using VevoStrain software system (VisualSonics Inc, Ontario, Canada).

## RNA extraction and sequencing

Total RNA was isolated from snap-frozen tissues using RNeasy Fibrous Tissue Mini Kit (Qiagen) for heart and skeletal muscle, RNeasy Mini Kit (Qiagen) for liver, or RNeasy Lipid Tissue Mini Kit (Qia-gen) for adipose tissue. cDNA was synthesized with the high-capacity cDNA Reverse Transcription Kit (Applied Biosystems) and random primers. mRNA expression of genes of interest was analyzed with TaqMan real-time PCR in a ViiA seven system (Applied Biosystems). RNA sequencing library were prepared with Illumina RNA-Seq with Poly-A selections. Subsequently, the libraries were sequenced on NovaSeq6000 (NovaSeq Control Software 1.6.0/RNA v3.4.4) with a 2 × 51 setup using 'NovaSeqXp' workflow in 'S1' mode flow cell. The Bcl was converted to FastQ by bcl2fastq_v2.19.1.403 from CASAVA software suite (Sanger/phred33/Illumina 1.8 + quality scale).

## RNA-sequencing data analysis

The raw RNA-sequencing results were processed using Kallisto (*Bray et al., 2016*) with index file generated from the Ensembl mouse reference genome (Release-96) (*Zerbino et al., 2018*). The output from Kallisto, both estimated count and TPM (Trancript per kilobase million), were subsequently mapped to gene using the mapping file retrieved from Ensembl BioMart website, by filtering only protein coding genes and transcripts. Genes with mean expression less than 1 TPM in each condition were filtered. For data exploration, we used PCA from sklearn package (*Pedregosa, 2011*) in Python 3.7 and used TPM values as the input.

Subsequently, we performed differential gene expression analysis using DESeq2 (*Love et al., 2014*) package in R. We utilized the capabilities from DESeq2 to normalize the rounded estimated count data and to correct for confounding factors (such as time). To define a gene as differentially expressed (DEGs), a gene has to fulfill a criterion of FDR < 5%. The results of differential expression analysis were then used for functional analysis.

We checked the tissue specificity of the DEGs in each tissue with the data from Mouse Gene Atlas (*Su et al., 2004*). For all the tissue-specific genes, we also checked their human-homolog genes in the human secretome database (*Uhlén et al., 2019*).

## Functional analysis

We performed functional analysis using the R package PIANO (*Väremo et al., 2013*). As the input, we used the fold changes and p-values from the DESeq2, and also GO BP and KEGG pathways gene-set collections from Enrichr (*Chen et al., 2013*; *Kuleshov et al., 2016*), and metabolites from Mouse Metabolic Reaction database (*Mardinoglu, 2015*). To define a process or pathway as significant, we used a cut off of FDR < 5% for the distinct direction of PIANO (both up and down).

## Co-expression network generation

We generated the co-expression network by generating gene-gene Spearman correlation ranks within a tissue type, using *spearmanr* function from SciPy (*Jones et al., 2001*) in Python 3.7. Using the same environment, we performed multiple hypothesis testing using Benjamini-Hochberg method from *statsmodels* (*Perktold et al., 2017*). Correlation data were filtered with criterion of adjusted p-value<5%.

The top 5% of filtered correlation results were then loaded into iGraph module (*Csardi and Nepusz, 2006*) in Python 3.7 as an unweighted network. To find the subnetworks, we employed the Leiden clustering algorithm (*Traag et al., 2019*) with *ModularityVertexPartition* method. Each cluster was analyzed by using Enrichr (*Chen et al., 2013*; *Kuleshov et al., 2016*) to get the enriched GO BP and KEGG pathways. Criterion FDR < 0.05 were used to find the significantly enriched terms. Clusters with less than 30 genes were discarded, to be able to get significant functional analysis results. Since GO BP was relatively sparse, we used Revigo (*Supek et al., 2011*) to summarize the GO BP into a higher level. Revigo was further employed to build a GO BP network. Clustering coefficient was calculated based on the average local clustering coefficient function within iGraph.

## Multi-tissue metabolic modeling

We combined tissue-specific models (of heart, liver, muscle, adipose and small intestine) constructed previously (*Mardinoglu, 2015*) in a multi-tissue model by adding an additional compartment representing the plasma, which allows the exchange of metabolites among different tissues. Blocked reactions that could not carry fluxes (and the unused metabolites and genes linked to these reactions) were removed from the models. In addition, the dietary input reactions and constraints were added to the small intestine model to simulate the food intake (*Supplementary file 7*). Specifically, we assumed that the mice weighed 30 g and consumed 4.5 g chow diet per day (15 g/100 g body weight) based on a previous study (*Kummitha et al., 2014*). We also calculated the tissue-specific resting energy expenditures and set them as mandatory metabolic constraints based on previous studies and resting energy expenditure for other tissues was incorporated by including a mandatory glucose secretion flux out from the system with the lower bound calculated based on ATP (*Supplementary file 7*; *Kummitha et al., 2014*).

To simulate the metabolic flux distribution in the sham-operated mice, we set the lipid droplet accumulation reaction in adipose tissue (m3_Adipose_LD_pool) as the objective function as we

assume the energy additional to the resting energy expenditure will be mostly stored as fat rather than used by the muscle for physical activities because mice raised in the cages might have very little exercise. Then, we used parsimonious FBA to calculate the flux distribution. To simulate the flux distribution after an MI, we calculated an expected flux fold change of each reaction based on the FDR and expression fold changes of all genes associated with the reaction, and obtains a flux distribution that is closest to this expected flux distribution while satisfying the stoichiometric balance and flux constraints of the model. The mathematical formulation of the method is described as below,

$$minimize\, Z = \sum_i \left| v_i - v_i^{exp} \right|$$

$$s.t. S * v = 0$$

$$lb \le v \le ub$$

where $S$, $v$, $lb$, $ub$ represent the stoichiometric matrix, flux distribution, lower bound and upper bound of all reactions, respectively. The $v_i^{exp}$ represents the expected flux of $i^{th}$ reaction which is calculated as follows,

$$v_i^{exp} = v_i^{ref} * \sqrt[n]{[m]} \prod_{j=1}^{m} FC_j$$

where $n$ is the number of gene sets that could independently catalyze the corresponding reaction, and $FC_j$ represents the expected expression changes of $j^{th}$ gene set which is calculated below,

$$FC_j = (1 - P_1) * fc_1 + P_1(1 - P_2) * fc_2 + \cdots + \prod_{k=1}^{m-1} P_k(1 - P_n) * fc_n$$

where $m$ is the number of genes in the $i^{th}$ gene sets, and $P_j$ and $fc_j$ respectively represents the FDR and fold change of gene expression with $j^{th}$ smallest fold change in this gene set. In this way, genes with lowest fold change will have a dominating effect within a gene set encoding a protein complex, while the geometric mean of expected fold changes of gene sets encoding different isozymes of this reaction will be used as the final expected flux fold change of this reaction.

### Validation of the results

We validated our findings by performing similar steps of RNA sequencing and functional analysis for the publicly available mouse MI datasets GSE104187 and GSE52313 (*Ounzain et al., 2015*; *Williams et al., 2018*).

### Data and code availability

All raw RNA-sequencing data generated from this study can be accessed through accession number GSE153485. Codes used during the analysis are available on https://github.com/sysmedicine/ArifEtAll_2020_MultiTissueMI (copy archeived at swh:1:rev:e79df3ef069674c1344c096ef6-b011e771cf506b; *Arif, 2021*).

## Acknowledgements

This work was financially supported by the Knut and Alice Wallenberg Foundation, Swedish Research Foundation and Swedish Heart-Lung Foundation.

## Additional information

### Competing interests

Maryam Clausen, Johannes Wikström, Damla Etal: employee at AstraZeneca. The other authors declare that no competing interests exist.

## Funding

| Funder | Grant reference number | Author |
| --- | --- | --- |
| Knut och Alice Wallenbergs Stiftelse | 72110 | Adil Mardinoglu Jan Boren |
| Vetenskapsrådet | | Jan Boren |
| Hjärt-Lungfonden | | Jan Boren |

The funders had no role in study design, data collection and interpretation, or the decision to submit the work for publication.

## Author contributions

Muhammad Arif, Conceptualization, Resources, Data curation, Software, Formal analysis, Investigation, Visualization, Methodology, Writing - original draft, Writing - review and editing; Martina Klevstig, Conceptualization, Data curation, Formal analysis, Investigation, Methodology, Writing - original draft, Writing - review and editing; Rui Benfeitas, Formal analysis, Validation, Methodology, Writing - review and editing; Stephen Doran, Formal analysis, Investigation, Methodology, Writing - review and editing; Hasan Turkez, Cheng Zhang, Writing - review and editing; Mathias Uhlén, Supervision, Writing - review and editing; Maryam Clausen, Johannes Wikström, Damla Etal, Conceptualization, Data curation, Writing - review and editing; Malin Levin, Investigation, Writing - review and editing; Adil Mardinoglu, Supervision, Funding acquisition, Writing - original draft, Project administration, Writing - review and editing; Jan Boren, Conceptualization, Supervision, Funding acquisition, Investigation, Writing - original draft, Project administration, Writing - review and editing

## Author ORCIDs

Muhammad Arif https://orcid.org/0000-0003-2261-0881
Rui Benfeitas http://orcid.org/0000-0001-7972-0083
Cheng Zhang http://orcid.org/0000-0002-3721-8586
Adil Mardinoglu https://orcid.org/0000-0002-4254-6090

## Ethics

Animal experimentation: This study were approved by the local animal ethics committee and conform to the guidelines from Directive 2010/63/EU of the European Parliament on the protection of animals used for scientific purposes.

## Decision letter and Author response

Decision letter https://doi.org/10.7554/eLife.66921.sa1
Author response https://doi.org/10.7554/eLife.66921.sa2

# Additional files

## Supplementary files

- Supplementary file 1. Differential expression analysis results.

- Supplementary file 2. KEGG pathways.

- Supplementary file 3. Gene ontology biological processes.

- Supplementary file 4. DEG comparison between liver and other tissues.

- Supplementary file 5. Reporter metabolite analysis.

- Supplementary file 6. Enrichment analyses of clusters, clusters properties.

- Supplementary file 7. Food intake, energy expenditure, and flux balance analysis (FBA) of whole-body modeling.

- Supplementary file 8. Validation result (differential expression and functional analysis).

- Supplementary file 9. Detailed information of 16 key genes that are DEGs in at least three tissues and neighbors and functional analysis results of The Neighbors of 4 key genes.

• Transparent reporting form

## Data availability

All raw RNA-sequencing data generated from this study can be accessed through accession number GSE153485.

The following dataset was generated:

| Author(s) | Year | Dataset title | Dataset URL | Database and Identifier |
|---|---|---|---|---|
| Arif M, Klevstig M, Benfeitas R, Doran S, Turkez H, Uhlén M, Wikström J, Zhang C, Levin M, Mardinoglu A, Boren J | 2021 | Integrative transcriptomic analysis of tissue-specific metabolic crosstalk after myocardial infarction | https://www.ncbi.nlm.nih.gov/geo/query/acc.cgi?acc=GSE153485 | NCBI Gene Expression Omnibus, GSE153485 |

The following previously published datasets were used:

| Author(s) | Year | Dataset title | Dataset URL | Database and Identifier |
|---|---|---|---|---|
| Williams AL, Khadka VS, Shohet RV | 2018 | Analysis of early changes in mouse heart transcriptome after myocardial infarction | https://www.ncbi.nlm.nih.gov/geo/query/acc.cgi?acc=GSE104187 | NCBI Gene Expression Omnibus, GSE104187 |
| Ounzain S, Micheletti R, Beckmann T, Schroen B, Alexanian M, Pezzuto I, Crippa S, Nemir M, Sarre A, Johnson R, Dauvillier J, Burdet F, Ibberson M, Guigo R, Xenarios I, Heymans S, Pedrazzini T | 2014 | Long RNA Seq on Sham vs MI Adult Mouse Heart | https://www.ncbi.nlm.nih.gov/geo/query/acc.cgi?acc=GSE52313 | NCBI Gene Expression Omnibus, GSE52313 |

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
