## [Decision Letter]

**Acceptance summary:**

This paper will be of considerable interest to researchers studying the interactions between metabolic responses in myocardial infarction. Ultimately this increased understanding of these metabolic responses could lead to exploration of new avenues of treatment.

**Decision letter after peer review:**

Thank you for submitting your article "Integrative transcriptomic analysis of tissue-specific metabolic crosstalk after myocardial infarction" for consideration by *eLife*. Your article has been reviewed by 3 peer reviewers, including Edward D Janus as the Reviewing Editor and Reviewer #1, and the evaluation has been overseen by a Senior Editor. The following individual involved in review of your submission has agreed to reveal their identity: Tunahan Cakir (Reviewer #2).

The reviewers and Editors have discussed their reviews with one another, and this letter is to help you prepare a revised submission.

Essential revisions:

1. RMetD2 formulation in the reference paper is complex/ difficult to follow. It seems RMetD2 predicts flux ranges rather than fluxes. If so, the definition in the paper is a bit oversimplification, and a more detailed definition should be made. Given that RMetD2 also uses differentially expressed genes to extract the model, it is actually not surprising that the results would match. So I still don't consider it to be an independent validation. RMetD2 in my opinion suffers from various drawbacks because it only uses differentially expressed genes to extract the network. Further, it also seems to strongly rely on fluxes rather than the structure of the genome-scale network. Whether fluxes correlate with expression, is still a hotly debated topic; especially for mammalian cells. Also, it appears that RMetD2 isn't published in any journal but only on biorxiv.

2. Sham operation – Could the authors please detail in the methods section what sham operation entails?

3. Could the authors show the coefficients of the first two principal components – to get an idea of how the gene space changes?

4. The colorbars are not clear throughout the manuscript.

5. The authors are making a model and predicting fluxes. Then, they performed a qualitative validation. Does the output in the section warrant all the effort going into building multi-tissue model? Its possible other computational methods could have done the job much easily. Could the authors show the necessity for building the multi-tissue model?

6. It is not clear to me what is purpose of the model? Models are built to capture the complexity of the problem. While the authors found a number of genes, it is not clear how these genes are producing complexity. The networks that the authors are using aren't clearly explained or delineated or benchmarked. Could the authors do some benchmarking and highlight the complexity of this network?

7. A lot of the analyses in the later part of the manuscript comes across as circular. The authors found some candidate genes implicated in this conditions using transcriptomics data, then they used transcriptomic data to find these DEGs? The validation was done the same way the initial part of the study was conducted. Isn't this biasing the study somehow? Typically, the data types used for validations are different than those for constructing the list of candidates. For e.g. when validating metabolic models of cells built using transcriptomic data, CRISPR-Cas9 essentiality screens are used. Here, they basically repeated the same analyses on the same transcriptome from a different experiment. What is the novel systems biology being learnt here?

8. "Functional analysis reveals widespread" seems problematic in terms of English.

---

## [Author Response]

Essential revisions:1. RMetD2 formulation in the reference paper is complex/ difficult to follow. It seems RMetD2 predicts flux ranges rather than fluxes. If so, the definition in the paper is a bit oversimplification, and a more detailed definition should be made. Given that RMetD2 also uses differentially expressed genes to extract the model, it is actually not surprising that the results would match. So I still don't consider it to be an independent validation. RMetD2 in my opinion suffers from various drawbacks because it only uses differentially expressed genes to extract the network. Further, it also seems to strongly rely on fluxes rather than the structure of the genome-scale network. Whether fluxes correlate with expression, is still a hotly debated topic; especially for mammalian cells. Also, it appears that RMetD2 isn't published in any journal but only on biorxiv.

We are sorry for the confusion here. We have used an updated version of RMetD2 where we only used the method to calculate the flux distribution in the MI condition and did not use it to extract the model. To avoid further confusion, we have described the updated method in detail in the revised manuscript as follows and do not refer to the outdated biorxiv version,

To simulate the flux distribution after an MI, we calculated an expected flux fold change of each reaction based on the FDR and expression fold changes of all genes associated with the reaction, and obtains a flux distribution that is closest to this expected flux distribution while satisfying the stoichiometric balance and flux constraints of the model. The mathematical formulation of the method is described as below,

minimizeZ=∑i|vi−viexp|s.t.S*v=0lb≤v≤ub where *S*, *v*, *lb*, *ub* represent the stoichiometric matrix, flux distribution, lower bound and upper bound of all reactions, respectively. The viexp represents the expected flux of *i*^th^ reaction which is calculated as follows, viexp=viref*∏j=1mFCjm where *n* is the number of gene sets that could independently catalyze the corresponding reaction, and FCj represents the expected expression changes of *j*^th^ gene set which is calculated below, FCj=(1−P1)*fc1+P1(1−P2)*fc2+⋯+∏k=1m−1Pk(1−Pn)*fcn where *m* is the number of genes in the *i*^th^ gene sets, and Pj and fcj respectively represents the FDR and fold change of gene expression with *j*^th^ smallest fold change in this gene set. In this way, genes with lowest fold change will have a dominating effect within a gene set encoding a protein complex, while the geometric mean of expected fold changes of gene sets encoding different isozymes of this reaction will be used as the final expected flux fold change of this reaction.

Even though we agree with the reviewer that fluxes might not perfectly correlate with gene expression, we think it will be equally, if not more, difficult to claim there is no correlation between these two. In addition, many previously published methods made similar assumptions as we did. As we only softly imposed the relative expression of the genes in the objective function and the final flux distribution is still largely decided by the stoichiometric structure and constraints of the model, we hope the reviewer will agree that the integration of the transcriptomic data in our model simulation is justified.

2. Sham operation – Could the authors please detail in the methods section what sham operation entails?

**Thank you for the comment. We have included an explanation about the sham operation procedure.**

**“Sham mice were handled identically (fasted, anesthetized, intubated, and connected to ventilator, and subsequently incised between 4^th^ and 5^th^ ribs), but no ligation of the LAD coronary artery was performed (and thus, no ischemia was induced in these mice).”**

3. Could the authors show the coefficients of the first two principal components – to get an idea of how the gene space changes?

Thank you for the comment. We have added the percentage of variance on each axis of the PCA to give more clarity on the transcriptional shifts (Related figure: Figure 1—figure supplement 1).

4. The colorbars are not clear throughout the manuscript.

Thank you for the comment. We have enlarged the color bars on each image to give more clarity to the figures.

5. The authors are making a model and predicting fluxes. Then, they performed a qualitative validation. Does the output in the section warrant all the effort going into building multi-tissue model? Its possible other computational methods could have done the job much easily. Could the authors show the necessity for building the multi-tissue model?

We would like to clarify that we performed the model simulation in this study to investigate the response to MI from a metabolic and holistic perspective. This is not the main focus of this study, but we believe this section provided complementary and valuable insight of the intra- and inter-tissues metabolic changes, such as the increased secretion of ketone body from adipose tissue. To address the reviewer’s concern, we extended this section to include more insight obtained from the metabolic modeling as follows:

“In addition, we observed an increase uptake of alpha-ketoglutarate (AKG) of heart 24h after MI. […] The influx of lactate has been reported to be positively correlated with the fraction of regional ejection of heart 36 and this net flux not only agrees well with the previous report but also additionally suggested the source of the lactate.”

6. It is not clear to me what is purpose of the model? Models are built to capture the complexity of the problem. While the authors found a number of genes, it is not clear how these genes are producing complexity. The networks that the authors are using aren't clearly explained or delineated or benchmarked. Could the authors do some benchmarking and highlight the complexity of this network?

We agree with the reviewer that models are platforms which could be used to capture the complexity of the problem. In this study, we performed multi-tissue metabolic modeling to mainly investigate the metabolic alteration and the inter-tissue metabolic cross-talking, and we did find some interesting results such as the elevated pentose phosphate pathway in heart as well as the increased ketone secretion from adipose tissue. We have substantially extended the result section of the metabolic modeling in the revised manuscript to include more insightful results from the metabolic modeling analysis.

Because the model we used in this study is based on a very unique data and we don’t know any studies which used similar mouse model, we enhanced the method section to clarify the model construction and flux simulation process in the revised manuscript as mentioned in other response considering this model is reconstructed mainly using well established tissue specific models that has been published previously.

7. A lot of the analyses in the later part of the manuscript comes across as circular. The authors found some candidate genes implicated in this conditions using transcriptomics data, then they used transcriptomic data to find these DEGs? The validation was done the same way the initial part of the study was conducted. Isn't this biasing the study somehow? Typically, the data types used for validations are different than those for constructing the list of candidates. For e.g. when validating metabolic models of cells built using transcriptomic data, CRISPR-Cas9 essentiality screens are used. Here, they basically repeated the same analyses on the same transcriptome from a different experiment. What is the novel systems biology being learnt here?

We would like to thank you for the constructive comments. We agree that we presented multiple transcriptomics analyses that have been used before. Apart from understanding the metabolic effect of MI in multiple tissues (which is unique as of now), our secondary goal is to propose a novel integrative framework for analyzing multi-tissue transcriptomics data based on the available techniques. We would like to emphasize that, even though the singular analyses were not novel, the integrative analysis in multi-tissue and disease setting both at transcriptomic and metabolic crosstalk level is a strong novelty of this study. This required employing not only state-of-art network analyses but also reconstruction of multi-tissue models through new methods that enable joint modeling of the metabolic interactions within and between tissues.

As this study is unique (as of now), we tried our best to validate it with other data with similar settings (from a tissue and we found only transcriptomics data) and run our pipeline to validate and strengthen our findings. Moreover, we also recognized the limitation that all the results presented in this study are purely based on transcriptomics data (as stated in the “Discussion” section of the manuscript). More experiments, such as with metabolomics and proteomics data, are in our pipeline to complement the results from the current study. In summary, we recognized the reviewer’s concerns and we would like to address it in our future studies.

8. "Functional analysis reveals widespread" seems problematic in terms of English.

Thank you for the constructive comments. We have adjusted the subtitle accordingly:

“Functional analysis reveals widespread alterations of mitochondrial, fatty acid, immune, and protein and RNA-related biological processes post MI with liver shows contrasting trend”.